Sequencing at sea: challenges and experiences in Ion Torrent PGM sequencing during the 2013 Southern Line Islands Research Expedition

Lim Yan Wei 1
Cuevas Daniel A. 2
Silva Genivaldo Gueiros Z. 2
Aguinaldo Kristen 1 3
Dinsdale Elizabeth A. 1
Haas Andreas F. 1
Hatay Mark 1
Sanchez Savannah E. 1
Wegley-Kelly Linda 1
Dutilh Bas E. 4 5
Harkins Timothy T. 6
Lee Clarence C. 6 7
Tom Warren 6
Sandin Stuart A. 8
Smith Jennifer E. 8
Zgliczynski Brian 8
Vermeij Mark J.A. 9 10
Rohwer Forest 1
Edwards Robert A. 1 5 11 raedwards@gmail.com
1 Department of Biology, San Diego State University , San Diego, CA , USA
2 Computational Sciences Research Center, San Diego State University , San Diego, CA , USA
3 Ion Torrent Research & Development Group, Thermo Fisher Scientific , Carlsbad, CA , USA
4 Centre for Molecular and Biomolecular Informatics, Radboud Institute for Molecular Life Sciences, Radboud University Medical Centre , Geert Grooteplein, GA, Nijmegen , The Netherlands
5 Department of Marine Biology, Institute of Biology, Federal University of Rio de Janeiro , Brazil
6 Advanced Applications Group, Life Technologies, Inc. , Beverly, MA , USA
7 Life Sciences Group, Thermo Fisher Scientific , South San Francisco, CA , USA
8 Center for Marine Biodiversity and Conservation, Scripps Institution of Oceanography, University of California San Diego , La Jolla, CA , USA
9 Caribbean Research and Management of Biodiversity (CARMABI) , Willemstad , Curacao
10 Aquatic Microbiology, Institute for Biodiversity and Ecosystem Dynamics, University of Amsterdam , Amsterdam , The Netherlands
11 Division of Mathematics and Computer Science, Argonne National Laboratory , Argonne, IL , USA
Souza Valeria
Electronic publication date: 2014 Aug 19
Publication date: 2014
Volume: 2
Electronic Location ID: e520
Received 2014 Jun 18; Accepted 2014 Jul 23
Copyright: © 2014 Lim et al.
Copyright year: 2014
Copyright holder: Lim et al.
License: This is an open access article distributed under the terms of the Creative Commons Attribution License, which permits unrestricted use, distribution, reproduction and adaptation in any medium and for any purpose provided that it is properly attributed. For attribution, the original author(s), title, publication source (PeerJ) and either DOI or URL of the article must be cited.
License URL: https://creativecommons.org/licenses/by/4.0/

Keywords: Genomics, Sequencing, Expeditions, Metagenomics, Environmental microbiology, Coral reef, Vibrio

Funding: NSF DEB-1046413 Gordon and Betty Moore Foundation GBMF-3781 CIFAR IMB-ROHW-141679 NSF CNS-1305112 NSF MCB-1330800 This work is partially supported by NSF Dimensions Grant (DEB-1046413; Edwards and Rohwer). This project was also funded in part by the Gordon and Betty Moore Foundation through Grant GBMF-3781 to Rohwer. Additional funding for Yan Wei Lim was provided by the Canadian Institute for Advanced Research (CIFAR; IMB-ROHW-141679). Additional funding for Edwards was provided by NSF grants CNS-1305112, and MCB-1330800. Dutilh was supported by an award from CAPES/BRASIL. The SDSU Vice President of Research, Director’s Office of Scripps Institution of Oceanography, Moore Family Foundation, and several private donors provided cruise support. The funders had no role in study design, data collection and analysis, decision to publish, or preparation of the manuscript.

==============================
Genomics and metagenomics have revolutionized our understanding of marine microbial ecology and the importance of microbes in global geochemical cycles. However, the process of DNA sequencing has always been an abstract extension of the research expedition, completed once the samples were returned to the laboratory. During the 2013 Southern Line Islands Research Expedition, we started the first effort to bring next generation sequencing to some of the most remote locations on our planet. We successfully sequenced twenty six marine microbial genomes, and two marine microbial metagenomes using the Ion Torrent PGM platform on the Merchant Yacht Hanse Explorer. Onboard sequence assembly, annotation, and analysis enabled us to investigate the role of the microbes in the coral reef ecology of these islands and atolls. This analysis identified phosphonate as an important phosphorous source for microbes growing in the Line Islands and reinforced the importance of L-serine in marine microbial ecosystems. Sequencing in the field allowed us to propose hypotheses and conduct experiments and further sampling based on the sequences generated. By eliminating the delay between sampling and sequencing, we enhanced the productivity of the research expedition. By overcoming the hurdles associated with sequencing on a boat in the middle of the Pacific Ocean we proved the flexibility of the sequencing, annotation, and analysis pipelines.

Introduction

DNA sequencing has revolutionized microbial ecology: next-generation sequencing has upended our traditional views of microbial communities, and enabled exploration of the microbial components of many unusual environments. In a typical environmental exploration, samples are collected at the study site, transported back to the laboratory, and analyzed after the scientific team returns from the field. This abstraction of the DNA sequencing from the sampling eliminates the possibility of immediate follow-up studies to explore interesting findings. In our previous studies of environmental microbial and viral components we identified questions and challenges that could have been answered with additional sample collection but awaited a future return to the field before they could be addressed (Dinsdale et al., 2008; Bruce et al., 2012).

The use of next-generation sequencing for microbial ecology involves two distinct components. First, the experimental aspects that include sample collection and preparation, DNA extraction, and sequencing, which are routine in the laboratory but challenging in the field. The principle limitation to taking sequencing into the field is the significant infrastructure and resources required for all steps of the sequencing process. In addition to the dedicated hardware required to generate the sequences, much of the hardware, and many of the sample preparation steps, require physically separated laboratory space (to reduce cross-contamination between the samples). Second, the informatics aspects, including processing the raw data into high quality sequences, comparing those sequences to existing databases to generate annotations, and the subsequent data analysis all require high-performance computational resources to generate meaningful biological interpretations (Meyer et al., 2008; Aziz et al., 2012; Edwards et al., 2012).

There are many challenges to next-generation sequencing in the field, but surmounting those obstacles will allow scientists to pursue new research avenues in exploring the environment using genetic approaches. Some of these challenges may be mitigated by the specific location being explored. For example, many terrestrial locations are accessible to mobile laboratories (Griffith, 1963; Grolla et al., 2011) and have access to cellular communications that can provide Internet access. Though low-bandwidth, these connections can be used for analysis of next-generation DNA sequences (Hoffman & Edwards, 2011). In contrast, aside from near-shore venues, Internet communication at remote marine stations typically relies on satellite transmissions, and thus is both limited in bandwidth and extremely expensive. New bioinformatics approaches are reducing the computational complexity of the algorithms in DNA sequence processing, therefore minimizing the resources needed for data analysis. Together with faster and cheaper computational technologies, these improved approaches mitigate the need for Internet-based computations (Edwards et al., 2012; Silva et al., 2014).

To explore the frontier of next-generation sequencing at sea, we deployed a Life Technologies Ion Torrent Personal Genome Machine (PGM) during the 2013 Southern Line Islands Research Expedition to sequence bacterial isolates and community metagenomes from these remote islands. We installed local bioinformatics capabilities to perform necessary sequence analysis. There were numerous challenges to remote DNA sequencing and analysis, however the end result—genome sequences generated at the remote central Pacific Atolls allowed us to focus our research on questions relevant to the samples we collected. In this paper we describe the sequencing and informatics pipelines established during the expedition, release the data generated during the 2013 Line Islands research expedition, and discuss some of the unexpected challenges in remote sequencing.

Methods

Study site

This study was performed during an expedition to the Southern Line Islands, central Pacific in October–November 2013 aboard the M/Y Hanse Explorer. Departing from Papeete Harbor, Tahiti, the islands visited were Flint (11.43°S, 151.82°W), Vostok (10.1°S, 152.38°W), Starbuck (5.62°S, 155.93°W), Malden (4.017°S, 154.93°W), and Millennium (previously called Caroline) (9.94°S, 150.21°W), in order, before returning to Papeete (Fig. S1).

Sample collections

Water samples were collected above the reef, and in-reef water samples were collected through crevices and against the benthos, both at 10 m depth. All sampling sites were named as either “tent sites” or “black reef”. “Supersucker” (Hass et al., in press) samples were collected from either coral or algal-surfaces with a modified syringe system which uses pre-filtered sterile seawater to flush the targeted microbial community from the respective surface (Fig. 1; Fig. S2). Metagenomics samples were collected from the benthic boundary layer of two sites at Starbuck islands; a newly discovered black reef site (Fig. S1; 5.62653°S, 155.90886°W) and the tent site (5.62891°S, 155.92529°W). The collection was performed using 19 l low density polyethylene collapsible bag (Cole-Parmer, Vernon Hills, IL, USA) connected to a modified bilge pump (Fig. 1) as we have described previously (Dinsdale et al., 2008). Large debris and eukaryotic cells were removed by filtration through 100 µm Nitex mesh and microbial cells were captured by passing the filtrate through the 0.45 µm Sterivex filter (Millipore, Inc., MA, USA). The Sterivex filters were stored at −20 °C until DNA extraction (Hass et al., in press).

Figure 1 Workflow for the preparation of bacterial isolates and water samples for genome and metagenome sequencing.

We started with sampling the coral reefs using supreme supersuckers, and isolates were cultured on different media. Isolates were assayed using multi-phenotype assay plates and sequenced.

This work was done under permit 012/13 from the Environment and Conservation Division of the Republic of Kiribati.

Bacterial isolates collection

A sample (100 µl) of each water sample was plated onto Thiosulfate-citrate-bile salts-sucrose (TCBS) agar for the isolation of Vibrio-like spp (Lotz, Tamplin & Rodrick, 1983). Typically >90% of colonies are Vibrio spp., but Pseudoalteromonas, Pseudovibrios, Shewanella and others also grow on TCBS. Therefore, colonies isolated from the TCBS plates were designated as Vibrio-like (F Thompson, pers. comm., 2014). In addition, a sample (100 µl) of each water sample was also plated onto Zobell’s Marine agar for the isolation of heterotrophic marine bacteria (ZoBell, 1941). In the naming scheme of isolates, “V” indicates Vibrio-like spp. and “Z” indicates isolates from Zobell’s Marine agar (Table 1). Single colonies were picked and re-streaked onto new agar plates for colony isolation. Vibrio-like isolates were selected based on the color and size of the colony. Non-vibrio isolates were selected based on the pigmentation (color) and colony morphology. Cells were scraped off the agar plate for DNA extraction, multi-phenotype assay plates (MAP), storage in RNA later, and metabolites extraction using 100% MeOH (Fig. 1). Permit regulations restrict the import and export of live biological material between Kritibati and the United States, and therefore viable bacteria are not available.

Table 1 Isolate characteristics.

Isolates and metagenomic sample information. The identifier of Vibrio spp. isolates are indicated by “V” and non-Vibrio spp. are indicated by “Z”. Potential genus was predicted based on the genome sequences.

Identifier	Island	Site (depth)	Potential genus	Culture media	Barcodea	
VRT1	Flint	Tent (10 m)	Vibrio	TCBS	1	
VRT2	Flint	Tent (10 m)	Vibrio	TCBS	2	
VRT3	Flint	Tent (10 m)	Vibrio	TCBS	3	
VRT4	Flint	Tent (10 m)	Vibrio	TCBS	4	
VAR1	Flint	Tent (10 m)	Vibrio	TCBS	5	
VAR2	Flint	Tent (10 m)	Vibrio	TCBS	6	
VAR3	Flint	Tent (10 m)	Vibrio	TCBS	7	
VAR4	Flint	Tent (10 m)	Vibrio	TCBS	8	
VRT5B	Flint	Tent–20 m	Vibrio	TCBS	8	
VRT11	Vostok	Supersucker (10 m - Coral)	Vibrio	TCBS	7	
VRT14	Vostok	In-reef (10 m)	Vibrio	TCBS	10	
VRT18	Starbuck	Ambient water (10 m)	Vibrio	TCBS	6	
VRT22	Starbuck	In-reef (10 m)	Vibrio	TCBS	3	
VRT23	Starbuck	In-reef (10 m)	Vibrio	TCBS	5 (4)	
VRT25	Starbuck	Black-reef water (10 m)	Vibrio	TCBS	4	
VRT30	Starbuck	Black-reef (surface)	Vibrio	TCBS	9	
VRT35	Malden	Supersucker (10 m - Coral)	Serratia	TCBS	11	
VRT37	Malden	Supersucker (10 m - Algae)	Serratia	TCBS	12	
VRT38	Millenium	Supersucker (10 m - Coral)	Vibrio	TCBS	13	
VRT41	Millenium	Supersucker (10 m - Algae)	Vibrio	TCBS	14	
ZRT1	Flint	Tent (10 m)	Pseudoalteromonas	Zobell	9 (3)	
ZRT3	Flint	Tent (10 m)	Pseudoalteromonas	Zobell	11	
ZAR1	Flint	Tent (10 m)	Pseudoalteromonas	Zobell	13 (1)	
ZAR2	Flint	Tent (10 m)	Ruegeria	Zobell	14 (2)	
ZRT28	Malden	Supersucker (10 m - Coral)	Serratia	Marine	16	
ZRT32	Malden	Supersucker (10 m - Algae)	Serratia	Marine	15	
SLI_3.1	Starbuck	Tent (10 m)	Mix	N/A	1	
SLI_3.2	Starbuck	Black-reef water (10 m)	Mix	N/A	2	
Notes.

a A new library was made and different barcode was used during the second library preparation.

DNA extraction and sequencing

The DNA from bacterial isolates was extracted and purified using the standard bacteria protocol in Nucleospin Tissue Kit (Macherey-Nagel, Dueren, Germany). In short, the cells were re-suspended with 180 µl T1 lysis buffer and mixed thoroughly. Proteinase K (25 µl) was added and the mixture was incubated at 37 °C for 3–8 h. The remaining extraction procedure was followed as recommended by the manufacturer protocol. Total microbial DNA was isolated from the Sterivex filters based on a modified protocol using the Nucleospin Tissue Kit (Macherey-Nagel, Dueren, Germany) (Kelly et al., 2012). Lysis steps were completed overnight at 37 °C in the Sterivex filters with double amount of Proteinase K-added T1 lysis buffer. An appropriate volume (200 µl for 180 µl T1 lysis buffer added, and 400 µl for 360 µl T1 lysis buffer added) of B3 lysis buffer was added for complete lysis before the lysate was removed from the Sterivex filter for subsequent extraction procedure as described in the manufacturers protocol. Sequence libraries were prepared using the Ion Xpress™ Plus Fragment Library Kit (Life Technologies, NY, USA) with slight protocol modification and each library is barcoded using the Ion Xpress™ Barcode Adapters 1–16 Kit. SPRI beads-based size selection according to the published New England Bioscience (NEB) E6270 protocol (https://www.neb.com/protocols/1/01/01/size-selection-e6270) was performed for 200–300 bp fragment size-selection after adapters ligation. Emulsion PCR was performed on 8-cycles amplified library using the OneTouch supplemented with Ion Torrent PGM Template OT2 200 Kit and template libraries were sequenced on the Ion Torrent PGM using the Ion Torrent PGM Sequencing 200 Kit v2 and Ion 318™ Chip Kit v2. Sequencing was performed across five different locations on the ship (Fig. 2).

Figure 2 A field guide in setting up sequencing workflow, specifically on a moving ship.

(A) Molecular bench for procedure including DNA extraction and library preparation was set-up in a clean room, which also serves as one of the bedrooms of scientists on-board. (B) Emulsion PCR was performed using the One-Touch technology located at the laundry room close to the hull of the ship. Damaged touch screen was replaced by re-wiring the display onto a laptop. (C) The sequencer was run at the owner’s cabin where there is an easy and safe access to the nitrogen tank that fuels the microfluidics of the sequencing technology. (D) A modified version of Ion Torrent server was set up to run the data analysis.

Multi-phenotype assay plate (MAP)

Bacterial cells were resuspended from single colonies into sterile artificial seawater. Before leaving San Diego, MAPs were created as stock plates using 48 different carbon substrates arrayed on the plate in duplicate (Table S1). Each stock well contains 1 ml of 6X basal media (6X MOPS media, 57 mM NH4Cl, 1.5 mM NaSO4, 30 µM CaCl2, 6 mM MgSO4, 1.9 MNaCl, 7.92 mM K2HPO4, 60 mM KCl, 36 µM FeCl3) and 1 ml of 5X carbon substrate. The substrates are used at a final concentration of 0.2% unless specified. Each experimental well on a 96-well plate consists of 50 µl of pre-mixed basal media + substrate solution, 75 µl sterile water, and 25 µl re-suspended bacterial cells. Bacteria cell optical density (OD) was read using spectrophotometer at 650 nm, at the start of the experiment (T = 0) and subsequently at the times noted. The multi-phenotype assay data were parsed and compiled using in-house PERL scripts (http://www.perl.org/). The data were visualized as growth curves by plotting OD measurements over time. Using the ggplot library in R (http://ggplot2.org/), the entire plate and curves were generated as images that were manually inspected (DA Cuevas, DR Garza, S Sanchez, JE Rostron, CS Henry, RA Overbeek, V Vonstein, F Rohwer, EA Dinsdale, RA Edwards, 2014, unpublished data). The OD measurements occurring at or after 40 h were extracted from the data for comparative analysis between the samples. These values were used to establish the 48 substrate vector profile of each sample. The Euclidean distance was calculated using the SciPy (Jones, Oliphant & Peterson, 2001) spatial distance module to generate a distance matrix that was the basis for a neighbor-joining tree (DA Cuevas, DR Garza, S Sanchez, JE Rostron, CS Henry, RA Overbeek, V Vonstein, F Rohwer, EA Dinsdale, RA Edwards, 2014, unpublished data). This code is available from GitHub at https://github.com/dacuevas/PMAnalyzer.

Bioinformatics analysis of sequence libraries

As noted below, the most common computational issue was corruption of the data files on the hard drives. To mitigate this issue, the MD5 checksum values for each file were calculated on the personal genome machine using the command line md5sum application. (The PGM contains a single hard drive.) This application was chosen because it is fast and efficient. The checksum for each file was computed and compared to the expected values before the computation started and at the completion of each computation.

On board ship, bases were called using a modified version of the Ion Torrent pipeline. To expedite the processing in the absence of a large compute cluster, the sequencing chip was digitally divided into four quadrants using the –cropped option to the bead finding application justBeadFind (part of the Ion Torrent suite, Life Technologies, Carlsbad, CA). A standard IonExpress 318 chip is 3,392 × 3,792 beads, and the chip was divided into four quadrants, 0–1,746, 0–1,946; 0–1,746, 1,846–3,792; 1,646–3,392, 0–1,946; and 1,646–3,392, 1,846–3,792. An overlap was provided on either side to ensure that all beads were identified. Any identical sequences from the same bead that was found in more than one quadrant were removed in post-processing steps. Bead finding, bead analysis, and base calling were performed using the Life Technologies software version 4.0.

The modified pipeline consisted of three steps with the following commands:

justBeadFind --cropped=0,0,1746,1946 --librarykey=TCAG --tfkey=ATCG --no-subdir --output-dir=sigproc sequences/ > justBeadFind.out 2> justBeadFind.err

Analysis --from-beadfind ../sequences > analysis.out 2> analysis.err

BaseCaller --trim-qual-cutoff 15 --trim-qual-window-size 30 --trim-adapter-cutoff 16 --input-dir ../sigproc > BaseCallerPost.hc.out 2> BaseCallerPost.hc.err

Sequences were separated based on the IonExpress primer sequence using a custom written PERL script that looked for exact matches to those primers (available from http://edwards-sdsu.cvs.sourceforge.net/viewvc/edwards-sdsu/bioinformatics/bin/split_sequences_by_tag.pl). Any sequences with a mismatch to the primers were ignored as potentially containing sequencing errors as has been proposed elsewhere (Schmieder & Edwards, 2011). Sequences shorter than 40 nt (including the tag) were discarded, and the tag sequence was removed prior to subsequent analyses. No other quality trimming was performed prior to assembly. Sequences were assembled using Newbler version 2.7 (Margulies et al., 2005), and scaffolds were constructed to closely related genomes using Scaffold Builder (Silva et al., 2013).

On the boat, contigs were annotated with a hybrid custom-written annotation pipeline based on both the RAST (Aziz et al., 2008; Overbeek et al., 2014) and the real time metagenomics system (Edwards et al., 2012). This pipeline uses amino acid k-mers to make functional assignments onto assembled DNA sequences, and extends the open reading frames (ORFs) containing the k-mer matches to identify the protein encoding regions. This heuristic pipeline eliminates largely overlapping ORFs but does not attempt to accurately identify the start positions of the genes, and is only concerned with those genes that can be functionally annotated from the k-mers. This approach was chosen as it is extremely fast, and provides a comprehensive annotation of the genes that can be identified in the database at the cost of the accuracy of exactly identifying the locations of the genes. This code is a part of the SEED toolkit and is available from http://biocvs.mcs.anl.gov/viewcvs.cgi/FigKernelScripts/assign_to_dna_using_kmers.pl. An online version of this is also available at the real time metagenomics site, http://edwards.sdsu.edu/RTMg/. In addition, tRNA-Scan SE was used to identify tRNA genes (Lowe & Eddy, 1997). Potential phage genes were identified by comparison to the PhAnToMe database (http://www.phantome.org/).

On return to shore, all assembled genomes were annotated using the standard RAST pipeline (Aziz et al., 2008; Overbeek et al., 2014) and metabolic models were built using the model seed (Henry et al., 2010).

Phylogenetic relationships of isolates

The 16S rRNA, rpoB, and recA gene sequences were extracted from the unassembled reads of each genome using the program genomePeek (K McNair, RA Edwards, 2014, unpublished data). Each group of sequences extracted from the same genome library were assembled into contigs using Newbler 2.7 (Margulies et al., 2005) with default parameters. The contigs were then grouped into 16S rRNA, RopB, and RecA gene group. Each group was aligned with ClustalW2 (Larkin et al., 2007) using the default parameters. The alignments were visually checked using Seaview (Galtier, Gouy & Gautier, 1996). Extraneous contigs were removed from the original set, and the remaining contigs were re-aligned, trimmed and exported in the PHYLIP format. Phylogenetic trees were generated using neighbor-joining clustering method (Larkin et al., 2007) and visualized using the interactive tree of life (Ciccarelli et al., 2006). This was not performed on the boat.

Genus designations

16S sequences were extracted as described above using Genome Peek, and the genus of the closest relative was used as the genus for the isolate. All GenomePeek analyses are available at http://edwards.sdsu.edu/GenomePeek/LineIslands/. This was performed on the boat, but was subsequently repeated.

Heat-map

A blastn search (Altschul et al., 1997) using an expected value cutoff of 10−5 was performed to compare all the 2013 expedition genomic data against 35 coral metagenomic data from previous expeditions (the metagenomic libraries vary from 996,778 bp to 117,975,100 bp). The portion of reads (just using the best hits) from each metagenomic sample that matched to each genome was calculated using Eq. (1) and implemented in a Python script available from http://edwards-sdsu.cvs.sourceforge.net/viewvc/edwards-sdsu/bioinformatics/LineIslandsGenomes/. (1) ∑i=1nAlignment_sizeiMetagenome_sizex∗Genome_sizey.

A heat-map was then generated to visualize the similarity between the 26 genomes sequenced on the 2013 expedition and the 35 sequenced coral metagenomes using an in-house Python script. This was initiated on the boat with a simple recruitment plot (available at the link above) and then subsequently refined.

Conserved functions

Function is best conserved between orthologous proteins (i.e., proteins that are derived from the same common ancestor). On the boat we determine which functions are conserved across all genomes by counting proteins that had the same annotation. This allowed us to quickly compare the common functions and identify genes that were unique to the strains that we sequenced. Upon our return, and after the RAST annotations we composed orthologous groups (OGs) specific for the organisms sequenced here. These orthologous groups represent protein families derived from a single protein in the common ancestor of the genomes and were identified by using a similar approach as previously described (Lucena et al., 2012). Briefly, we first queried the complete proteomes using an all-by-all blastp search (Altschul et al., 1997). The resulting bitscores were used to define in-paralogous groups of recently duplicated genes (i.e., after the last speciation event) within every genome. Within the genome, all proteins with a matching score better or equal than to any protein in another genome were joined into an “in-paralogous group”. We then combined the in-paralogous groups conservatively between species by joining pairs of reciprocal best blastp hits to create the final list of orthologous groups for the complete set of genomes.

Identification of genes required for growth on L-serine

The microbes were scored for growth on L-serine in the multi-phenotype assay plates. A matrix was constructed listing all the genomes and all the functional roles annotated as being present in those genomes, with the values in the matrix being whether each functional role was present in each genome. Two approaches were used to identify those genes that separate the strains that can grow on L-serine as a sole carbon source from those strains that can not. First, a random forest machine learning approach was used (Breiman, 2001), with the genes as variables and the ability to grow on L-serine as categories. The random forest identifies important variables (genes) that discriminate the two categories (Dinsdale et al., 2013). The approach used the R package “randomForest” (Liaw & Wiener, 2002) to classify the matrix. Second, a simple summation approach was used, counting the number of organisms that contained each gene that could or could not utilize L-serine. This table was sorted to identify those genes that are present in the strains that can utilize L-serine and absent from those strains that could not utilize L-serine. Both approaches gave similar results. This was initiated on the boat (with the random forest approach) and then subsequently refined.

Results

During the 2013 Southern Line Islands research expedition we deployed a next-generation sequencing instrument, isolated DNA from bacteria and metagenomes, sequenced the DNA and analyzed the samples. This is the first time that next-generation sequencing has been used in remote field locations. We demonstrated that we can deploy a next-generation sequencer successfully wherever microbial ecology is being studied.

Genome sequencing

Solely using the Ion Torrent PGM sequencing technology (Life Technologies), twenty six genomes and two metagenomes were successfully sequenced onboard the M/Y Hanse Explorer during the three weeks expedition in Southern Line Islands (Fig. S1; Table 1). We generated close to 1.5 billion bases (post quality filtering) of high quality DNA sequence data to investigate the role of microbes on the world most pristine coral reef ecosystems. Additionally, more than 7.5 billion bases (post quality filtering) were generated by Life Technologies to supplement the dataset with additional six genomes (Table 2) from the last two islands and to increase the amount of data of those under-sequenced libraries. In total, we sequenced three Pseudoalteromonas; one Ruegeria; two Serratia; and twenty Vibrio isolates. All sequences have been deposited in public databases (Table 3). On board, culturing, DNA extraction, library construction, and sequencing took approximately 5 days to complete. The analysis of the sequence chips took approximately 5 h per quadrant to complete, and the annotation of those sequences took about an hour to complete. As noted below, analysis of the genomes and metagenomes was the most time consuming part of the bioinformatics analysis, and remains an ongoing project.

Table 2 Sequencing data.

Library characteristics of the genomes and metagenomes.

Identifier	Total reads	Total bases (bp)	Contigs (>1 kbp)	Longest contig (bp)	Size (Mbp)	
VRT1	370,046	67,150,180	784	46,040	4.97	
VRT2	873,103	163,538,506	320	169,085	5.73	
VRT3	437,353	81,088,282	771	54,972	5.61	
VRT4	432,329	80,084,392	626	45,526	5.10	
VAR1	305,860	55,268,349	1,549	20,978	5.58	
VAR2	651,865	114,981,090	463	67,906	4.95	
VAR3	553,761	102,526,008	2,715	15,535	5.71	
VAR4	415,376	71,395,671	1,386	31,755	5.65	
VRT5B	446,202	68,187,702	1,624	28,687	4.76	
VRT11	296,834	50,764,788	1,219	23,083	4.45	
VRT14	591,693	105,091,668	612	94,938	5.52	
VRT18	407,884	71,662,502	1,330	43,832	5.66	
VRT22	500,878	79,757,251	1,506	27,655	5.53	
VRT23b	1,913,266	321,123,402	34	994,789	5.62	
VRT25	237,978	35,973,495	1,655	7,836	2.88	
VRT30	1,135,838	195,189,151	1,872	34,535	5.5	
VRT35a	5,074,872	836,924,832	1,227	157,763	12.27	
VRT37a	3,505,390	602,639,979	191	172,237	5.16	
VRT38a	4,739,179	852,206,121	47	742,807	5.97	
VRT41a	5,189,926	807,748,928	40	1,680,777	5.76	
ZRT1b	5,927,108	885,069,802	104	413,587	5.78	
ZRT3	515,406	76,920,334	1,039	29,051	5.07	
ZAR1b	3,127,071	289,378,345	423	111,609	5.30	
ZAR2b	2,975,125	269,722,069	2,380	50,061	7.18	
ZRT28a	6,114,713	1,110,443,467	113	263,705	5.16	
ZRT32a	4,932,760	903,726,978	900	214,396	6.28	
SLI_3.1b	5,381,011	665,058,375	–	–	–	
SLI_3.2	186,516	26,325,947	–	–	–	
Total	9,093,303,263				
Notes.

a Libraries sequenced by Life Technologies after the expedition.

b These libraries were under-sequenced during the expedition. Additional sequence data was provided by Life Technologies.

Table 3 Accession numbers.

Data accession numbers for the sequences. All sequences are in NCBI bioproject PRJNA253472.

Sample
ID	RAST
ID	NCBI
sample ID	NCBI
Taxonomy ID	NCBI locus
tag	NCBI SRA
sample	NCBI SRA
experiment	NCBI SRA
run	
SLI-3.1		SAMN02904893	408172	–	SRS655288	SRX648027	SRR1509098	
SLI-3.2		SAMN02904892	408172	–	SRS655289	SRX648028	SRR1509099	
VAR1	163649	SAMN02870694	1515472	HQ00	SRS655290	SRX648029	SRR1509100	
VAR2	163650	SAMN02870695	1515473	HQ01	SRS655291	SRX648030	SRR1509101	
VAR3	163651	SAMN02870696	1515474	HQ02	SRS655292	SRX648031	SRR1509102	
VAR4	163652	SAMN02870697	1515475	HQ03	SRS655293	SRX648032	SRR1509103	
VRT1	165046	SAMN02870670	1519375	HM87	SRS655294	SRX648033	SRR1509104	
VRT11	163654	SAMN02870717	1515495	HQ22	SRS655295	SRX648034	SRR1509105	
VRT14	163656	SAMN02870718	1515496	HQ23	SRS655296	SRX648035	SRR1509106	
VRT18	163657	SAMN02870712	1515490	HQ17	SRS655297	SRX648036	SRR1509107	
VRT2	163646	SAMN02870698	1515476	HQ04	SRS655298	SRX648037	SRR1509108	
VRT22	163658	SAMN02870713	1515491	HQ18	SRS655299	SRX648038	SRR1509109	
VRT23	163659	SAMN02870714	1515492	HQ19	SRS655300	SRX648039	SRR1509110	
VRT25	163660	SAMN02870715	1515493	HQ20	SRS655301	SRX648040	SRR1509111	
VRT3	163647	SAMN02870699	1515477	HQ05	SRS655302	SRX648041	SRR1509112	
VRT30	163661	SAMN02870716	1515494	HQ21	SRS655303	SRX648042	SRR1509113	
VRT35	163662	SAMN02870706	1515484	HQ11	SRS655304	SRX648043	SRR1509114	
VRT37	163663	SAMN02870707	1515485	HQ12	SRS655305	SRX648044	SRR1509115	
VRT38	163664	SAMN02870710	1515488	HQ15	SRS655306	SRX648045	SRR1509116	
VRT4	163648	SAMN02870700	1515478	HQ06	SRS655307	SRX648046	SRR1509117	
VRT41	163665	SAMN02870711	1515489	HQ16	SRS655308	SRX648047	SRR1509118	
VRT5B	163653	SAMN02870701	1515479	HQ07	SRS655309	SRX648048	SRR1509119	
ZAR1	163669	SAMN02870702	1515480	HQ08	SRS655310	SRX648049	SRR1509120	
ZAR2	163670	SAMN02870703	1515481	HR57	SRS655311	SRX648050	SRR1509121	
ZRT1	163666	SAMN02870704	1515482	HQ09	SRS655312	SRX648051	SRR1509122	
ZRT28	163671	SAMN02870708	1515486	HQ13	SRS655313	SRX648052	SRR1509123	
ZRT3	163667	SAMN02870705	1515483	HQ10	SRS655314	SRX648053	SRR1509124	
ZRT32	163672	SAMN02870709	1515487	HQ14	SRS655315	SRX648054	SRR1509125	

Twenty Vibrio isolates corresponding to five Vibrio spp. including V. harveyi (and potentially its sister species, V. campbellii), V. coralliitycus, V. alginolyticus, V. shilonii, and V. cyclitrophicus were cultured and their genomes were sequenced. Non-Vibrio isolates whose genomes were sequenced included Pseudomonas fluorescence, Serratia proteamaculans, Serratia marcescens, Pseudoalteromonas spp., and Phaeobacter gallaeciensis. Sequencing these genomes with Ion Torrent PGM demonstrated that approximately 1 gigabase (109 bp) of DNA sequence is required to assemble typical marine microbial genomes to less than 100 contigs using this technology (Fig. S3). The quality of assembly for the genomes appears to be solely dependent on the number of reads generated, and thus with sufficient time and resources all the genomes could be reduced to less than 100 contigs (high-quality draft status).

These genomes were annotated onboard the Hanse Explorer using our rapid annotation pipeline. Based on these annotations, the ten closest genomes to our newly sequenced genomes were identified, and the presence and absence of genes in those genomes summarized to identify the unique functions in our genomes. Subsequently, we also created groups of orthologous genes to identify those genes unique to our isolates. In total we identified 11,585 orthologous groups in the genomes. Each genome had 3,032 ± 550 orthologous groups. There were 1,442 orthologous groups that were unique to the Vibrio-like genomes and 4,913 orthologous groups that were unique to the Zobell genomes (see Table S2). Presumably these are the specialization genes that allow these organisms to grow on the reefs of the Southern Line Islands. Many of these genes are things that have been identified previously as separating microbial species, such as prophages (Akhter, Aziz & Edwards, 2012), transposons (Aziz, Breitbart & Edwards, 2010), IS elements and other mobile genes (Edwards, Olsen & Maloy, 2002).

All of the sequenced isolates contain prophage-like elements, suggesting phage predation controls bacterial populations as we have shown before (Dinsdale et al., 2008). Many of the genomes also contained nucleases and CRISPR elements indicative of resistance to active phage infections. The bacteria may be responding to phage infections by altering their cell surface, and genes involved in alternative pathways to construct lipopolysaccharide (LPS) were unique to some of the strains that we sequenced. Twenty of the twenty-six genomes contain variable genes involved in the synthesis of β-L-rhamnose, a deoxy-sugar that that is a building block of LPS (Kanehisa et al., 2004). Some examples include glucose-1-phosphate thymidylyltransferase similar to E. coli rfbA; dTDP-glucose 4,6-dehydratase similar to E. coli rfbB; dTDP-4-dehydrorhamnose 3,5-epimerase similar to E. coli rfbC; dTDP-4-dehydrorhamnose reductase similar to E. coli rfbD. In the isolate VRT11, for example, these four genes are located adjacent to each other, presumably in a single operon within the rfb gene cluster.

Phosphorous is essential for growth but is often limiting in marine environments since most phosphate salts are insoluble (Stanier, Adelberg & Ingraham, 1979). Phosphorous is readily converted to phosphonates, compounds that contain C–P bonds (rather than the more typical C–O–P bonds of phosphates) by the phosphoenolpyruvate mutase (PepM) mediated isomerization of phosphoenolpyruvate to phosphonopyruvate (Yu et al., 2013). In marine environments, phosphonate production is catalyzed by Prochlorococcus and Pelagibacter, but is also catalyzed by marine mollusks, anemones and by members of the coral holobiont (Thomas et al., 2009; Yu et al., 2013). Phosphonate utilization by Vibrio species has been shown in mescosm experiments using surface water of the North Pacific Subtropical Gyre (Martinez et al., 2013). However, not all Vibrio isolates are able to utilize phosphonate. For example, the coral pathogen V. shiloi AK1 (Kushmaro et al., 2001) is predicted to be able to use phosphonate, while the coral pathogen V. coralliilyticus (Ben-Haim et al., 2003) is not able to use phosphonate. Eighteen of the isolates that were sequenced here (VAR3, VAR4, VRT2, VRT4, VRT5B, VRT14, VRT22, VRT23, VRT25, VRT35, VRT37, VRT3, VRT41, ZAR1, ZAR2, ZRT3, ZRT28, and ZRT32) contained phosphonate transporters and utilization genes, suggesting that in the oligotrophic waters of the Southern Line Islands, phosphonate is a critical phosphorous source for heterotrophic bacteria and they likely scavenge it from the coral reef.

Iron is also often limiting in offshore marine environments in the Southern Ocean (Martin, 1992), and the exogenous addition of iron to reef systems (e.g., from ship wrecks) promotes the over-growth of algae (Kelly et al., 2012). The presence of a multitude of iron acquisition mechanisms, including high affinity transporters for both ferric (Fe3+) and ferrous (Fe2+) iron, ABC transporters, and an average of twenty siderophore genes per genome suggests that the marine isolates from the Southern Line Islands actively scavenge iron and are poised to consume any additional iron that enters the system.

Phenotypic analysis

In addition to sequencing the genomes of all isolates, we examined the phenotypic differences by using a multi-phenotype assay plate. The MAP allowed us to quantitatively measure the cellular phenotypes of each isolate in response to different nutrient sources based on their growth. Examples of the growth curves for all 48 carbon sources are shown for isolates VRT1 and VRT2 (Fig. S4). The growth characteristics of each isolate in the 48 carbon sources used in this experiment are shown in Fig. S5 as a heatmap. The growth curves from the negative controls and filtered seawater-only samples displayed no change in OD650 over time (Fig. S6); the OD650 measurement was consistently below 0.10 in those controls indicating a viable protocol and setup.

Although a few isolates (e.g., ZAR1, VAR2, and VAR4) were only able to grow on a few compounds, most of the isolates were generalists, able to grow on a wide range of carbon and nitrogen sources (Fig. S5). The isolates did not separate by island of isolation, suggesting that any variations in oceanographic conditions among the atolls are outweighed by biological influences (see below).

Serine utilization

Free serine is abundant in the ocean and we previously proposed that serine is used as an osmolyte by marine microbes (Rodriguez-Brito, Rohwer & Edwards, 2006). Fifteen of the twenty-six isolates that we assayed were able to grow on serine as a sole carbon source (VAR3, VRT1, VRT3, VRT4, VRT5B, VRT14, VRT18, VRT22, VRT23, VRT30, VRT35, VRT38, VRT41, ZAR2, and ZRT1), and we therefore examined which genotypes are responsible for growth on serine. L-serine dehydratase (E.C. 4.3.1.17), the enzyme that converts L-serine to pyruvate and ammonia, is in every one of the genomes that we sequenced except VAR3, and is almost always associated with a serine transporter (including in all of those strains that can not utilize serine as a sole carbon source). D-serine dehydratase (E.C. 4.3.1.18) that performs the same reaction with D-serine is in twenty of the genomes that we sequenced. We therefore compared the features present in the genomes to identify which annotations are associated with serine catabolism. Genes involved in vitamin B12 synthesis (cobalamin; cobU, cobS) and the conversion of serine to homocysteine (O-acetylhomoserine sulfhydrylase (EC 2.5.1.49)/O-succinylhomoserine sulfhydrylase (EC 2.5.1.48)) are present in all of the strains that can use L-serine as a sole carbon source and few of those strains that cannot. These enzymes all connect serine catabolism to methionine metabolism via homocysteine (so that S-adenosyl methionine that catalyzes the reaction can be replaced). It has previously been shown in E. coli that growth with L-serine as a sole carbon source is dependent on methionine metabolism (Brown, D’Ari & Newman, 1990) suggesting that in marine microbes a similar requirement holds and these microbes are using the same metabolic pathways.

Comparison to metagenomes

Following our previous expeditions to the Line Islands we sequenced 33 microbial metagenomes, and during the most recent expedition we sequenced two additional metagenomes. As shown in Fig. 3, comparing the microbial genomes that we sequenced with the microbial metagenomes shows that we have observed each of the microbial genomes previously in our metagenomic sequences. Similarity between the genomes and metagenomes was not dependent on either the metagenome size, genome size, or sequence coverage. ZAR2, the unique Ruggeria, and ZRT1, a Pseudoalteromonas, have unique profiles when compared to the metagenomes. This suggests that these organisms may be either transient colonizers of the reef that are passing through, or low abundance colonizers that are rarely sampled and we isolated them by chance. In contrast, the Serratia and most of the Vibrio clones are frequently found in the different samples and are therefore likely generalists. However, the uniform similarity across genera (Vibrio, Pseudoalteromonas, and Serratia) suggests that the previous metagenome sequences contain the genus- and species-specific genes (e.g., housekeeping genes) of those organisms and not necessarily the strain-specific genes that may be unique in these organisms (Kelly et al., 2014).

Figure 3 Heatmap comparing all genomes sequenced to Line Islands metagenomes.

The genomes, along the horizontal axis, are clustered by genera. The metagenomes on the vertical axis are organized by the island from which the samples were isolated. Solid gray bars on the top indicate the assembled genome size (bp), and gray bars on the right indicate metagenome size (bp). The cells are colored by distance calculated as described in the text (Eq. (1)) and as shown in the legend on the lower left.

Discussion

Next-generation sequencing has revolutionized microbial ecology but has always remained a step away from the field work. Samples are collected, returned to the lab, and studied. In many ways this is analogous to the field ecologists of the 19th Century that captured wild beasts and brought them to zoos or museums to study. There are many reasons why researchers will want to use sequencing in remote locations, including the limitations of permitting, archiving samples, and the ability to perform experiments as soon as the data is generated. With the advancements in next-generation sequencing, sample preparation, and data analysis, microbes can be studied in their natural habitat. By bringing the instruments to the environment, and not the other way around, environmental microbiology can be explored in heretofore unimagined ways.

The onboard sequencing and analysis suggested that microbes in the Southern Line Islands are limited by phosphorous and iron. The genomes predicted their potential to scavenge phosphorus from phosphonate, and iron from a variety of sources through various transporters and siderophores. Approximately half of the microbes that were isolated are able to grow on L-serine by converting L-serine to methionine. Although there does not appear to be any genus-specific preference for growing on L-serine (some isolates of Vibrio, Pseudoalteromonas, and Serrtatia could grow on L-serine), there is a specific biochemical pathway that is required: the transformation of L-serine to methionine via cobalamine. We could not identify any correlation between the ability to utilize serine and the location where the microbes were isolated, at any scale from kilometers to micrometers. It therefore remains to be determined what selects for the ability to utilize L-serine as a sole carbon source in the marine environment.

The physical distance between the five islands is shown in Fig. S1. Island biogeography suggests that closer islands should have more related organisms (MacArthur & Wilson, 2001). To test whether the microbes on the Southern Line Islands follow this rule, we calculated genetic distance between each of the isolates based on several marker genes (16S, rpoB, and recA), the genotypic distance based on the presence of orthologous groups in each genome, and the phenotypic distance based on the multi-phenotype assay plates (Fig. 4). There was no correlation between the distance between the islands and the genetic, genotypic, or phenotypic distances, suggesting that the microbes of the Southern Line Islands are not constrained to their local islands and are not restricted in their migration between islands (Fig. S7).

Figure 4 Distance tree comparing isolates based on genotypes and phenotypes.

(A) Distance tree based on the growth of isolates on 48 different carbon sources, using the optical density measurement at 40 h. (B–D) Distance tree based on the 16S rRNA (B), RecA (C), and RpoB (D) genes extracted from each genome.

Challenges with onboard sequencing

The first challenge to sequencing on a boat was organizing the equipment to minimize the possibility of cross-contamination between the samples. The hardest part of the microbiology and molecular biology was keeping everything clean. On the M/Y Hanse Explorer, the microbiology lab was on the upper aft deck, the DNA isolation and quantification station was in a cabin, and the PCR station was in the dining room (Fig. 2). Because the OneTouch contains a centrifuge, this equipment was placed in the lowest part of the ship, the laundry room, for maximum stability. The Ion Torrent PGM was housed in the owner’s quarters, atop the ship, to allow connection to the nitrogen gas tank which was stored outside. Centrifugation poses a significant problem on boats because of the conservation of angular momentum. Therefore, whenever possible, centrifugation was eliminated from the protocol. For example, cultures were grown to dense colonies to avoid the need to pellet cells, a mini-centrifuge was used for column based DNA extraction, and vacuum-based purification protocols were used as a back up.

The computational aspects were surprisingly challenging. First, there were the unexpected equipment failures that had to be overcome without access to technical support or replacements. The touch screen on the OneTouch did not survive transit to the vessel, and control of that instrument had to be reverse engineered using the X11 interface and a Linux laptop. Second, data analysis requires consistent read/write access to the disks, and that process frequently experienced data corruption on compute server and resulted in the potential for loss of data. The solution that was implemented was to compute the md5sum (essentially a unique string that represents the size and contents of the file) for each file on the Ion Torrent PGM hard drive, and continually compare the md5sum calculated for the files on the compute server with those on the Ion Torrent PGM hard drive. Any deviation in the calculated values suggested that the file had been corrupted. It is not known what caused the data corruption as upon returning to San Diego the server has been through several compute cycles without a file corruption. We speculate that it was most likely the motion of the boat (as noted above for centrifugation) or potentially the uneven power that is available on a ship. It is likely that implementation of a RAID system in the PGM or the substitution of the hard drive with a solid-state drive would also mitigate these issues. The third problem that had to be overcome was ensuring appropriate compute resources for data analysis. As discussed in methods, the Ion Torrent PGM data files are amenable to partial processing, which reduces the memory footprint and computational time required to analyze the data. The final problem is to ensure that there is sufficient expertise available to analyze the data in a timely manner. Two proposed solutions include enabling all members of the scientific team access to the data via a local (ship-board) Wi-Fi or sending the data off the ship for remote analysis. The latter is potentially feasible as sequence data is highly compressible and thus resource requirements for data transfer can be reduced.

Conclusion

This was the first successful attempt to bringing next-generation sequencing into a remote field expedition. We sequenced twenty six bacterial genomes and two metagenomes. The real-time analyses of these data unearth unique metabolic processes that contribute to their survival in the Southern Line Islands.

Supplemental Information

Table S1 Media formulations in phenotype assay plates

In-house phenotypic array plate. Each stock well contains 1 ml of 6X basal media (6X MOPS media, 57 mM NH4Cl, 1.5 mM NaSO4, 30 µM CaCl2, 6 mM MgSO4, 1.9 M NaCl, 7.92 mM K2HPO4, 60 mM KCl, 36 µM FeCl3) and 1 ml of 5×carbon substrate. The substrates are used at a final concentration of 0.2% unless specified.

Click here for additional data file.

Table S2 Orthologous groups in all the genomes

Orthologous groups identified in all the genomes. OG is a unique identifier; Function is the most common function annotated for all the proteins; The remaining columns identify the protein encoding genes (pegs) found in each genome or None if the gene is not present.

Click here for additional data file.

Figure S1 2013 Southern Line Islands Research Expedition

Expedition route of the 2013 Southern Line Island Expedition. The expedition left Papeete Harbor and circumnavigated the Southern Line Islands in a clockwise direction.

Click here for additional data file.

Figure S2 Sampling the reefs

Examples of sampling surface for bacterial isolates. VRT35 and VRT38 were isolated from coral surfaces, while VRT37 and VRT41 were isolated from algae surfaces.

Click here for additional data file.

Figure S3 Number of bases sequenced and number of contigs generated

Number of bases sequenced and number of contigs generated. The best-fit power regression of the data (shown with a gray line) has the equation [number of contigs] = 3.87∗1010 [base piars sequenced]−0.95 with an R2 of 0.667.

Click here for additional data file.

Figure S4 Growth curves in 48 different media showing growth in some wells

Growth curve of bacterial isolates in 48 different carbon sources indicated in Table S1.

Click here for additional data file.

Figure S5 Heat map showing growth characteristics of all the isolates

Heatmap showing the growth characteristics of each bacterial isolates from the 2013 Southern Line Island expedition in 48 different carbon sources.

Click here for additional data file.

Figure S6 Negative controls don’t grow

Growth curve of negative controls and filtered seawater-only samples in 48 different carbon sources indicated in Table S1.

Click here for additional data file.

Figure S7 There is no correlation between genetic and physical distance

Correlation between physical distance and genetic distance in the microbial species purified on the Line Islands.

Click here for additional data file.

A special thank you to the Captain, Martin Graser, and crew of the M/Y Hanse Explorer for their assistance during the expedition. Dive and general common sense was provided by Christian McDonald (SIO). We thank Life Technologies for providing library preparation and sequencing reagents for this study. We thank Marina Kalyuzhnaya for helpful and enlightening discussions about serine and methionine metabolism.

Additional Information and Declarations

Competing Interests

Author Contributions

Field Study Permissions

DNA Deposition

Data Deposition

Kristen Aguinaldo, Timothy Harkins, Clarence Lee, and Warren Tom are employees of Life Technologies, Inc. and Mark J.A. Vermeij is an employee of the Caribbean Research and Management of Biodiversity (CARMABI).

Yan Wei Lim and Robert A. Edwards conceived and designed the experiments, performed the experiments, analyzed the data, wrote the paper, prepared figures and/or tables, reviewed drafts of the paper.

Daniel A. Cuevas and Genivaldo Gueiros Z. Silva analyzed the data, wrote the paper, prepared figures and/or tables, reviewed drafts of the paper.

Kristen Aguinaldo analyzed the data, contributed reagents/materials/analysis tools, reviewed drafts of the paper.

Elizabeth A. Dinsdale, Savannah E. Sanchez, Timothy T. Harkins, Clarence C. Lee and Warren Tom contributed reagents/materials/analysis tools, reviewed drafts of the paper.

Andreas F. Haas performed the experiments, analyzed the data, reviewed drafts of the paper.

Mark Hatay contributed reagents/materials/analysis tools, prepared figures and/or tables, reviewed drafts of the paper.

Linda Wegley-Kelly and Bas E. Dutilh analyzed the data, reviewed drafts of the paper.

Stuart A. Sandin, Jennifer E. Smith and Brian Zgliczynski reviewed drafts of the paper, organized the field expedition, permits, and logistics.

Mark J.A. Vermeij performed the experiments, reviewed drafts of the paper.

Forest Rohwer conceived and designed the experiments, performed the experiments, reviewed drafts of the paper.

The following information was supplied relating to field study approvals (i.e., approving body and any reference numbers):

All samples were collected on SCUBA and under the Scientific Research Permit 021/13 issued by the Environment and Conservation Division of the Republic of Kiribati.

The following information was supplied regarding the deposition of DNA sequences:

Sequences have been deposited in RAST and GenBank and accession numbers are provided in Table 3.

The following information was supplied regarding the deposition of related data:

Figures are available at FigShare:

F1: http://dx.doi.org/10.6084/m9.figshare.1130904;

F2: http://dx.doi.org/10.6084/m9.figshare.1130905;

F3: http://dx.doi.org/10.6084/m9.figshare.1130906;

F4: http://dx.doi.org/10.6084/m9.figshare.1130914;

S1: http://dx.doi.org/10.6084/m9.figshare.1130907;

S2: http://dx.doi.org/10.6084/m9.figshare.1130908;

S3: http://dx.doi.org/10.6084/m9.figshare.1130909;

S4: http://dx.doi.org/10.6084/m9.figshare.1130910;

S5: http://dx.doi.org/10.6084/m9.figshare.1130911;

S6: http://dx.doi.org/10.6084/m9.figshare.1130912;

S7: http://dx.doi.org/10.6084/m9.figshare.1130913.

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
