# Peer review of "Sequencing at sea: challenges and experiences in Ion Torrent PGM sequencing during the 2013 Southern Line Islands Research Expedition"

_PeerJ, doi:10.7717/peerj.520_

## Round 0.1 · original submission · Minor Revisions

This is a very well done study, however, please follow both reviewers recommendations.

·

Basic reporting

This paper is clearly written and well organized. The introduction and background are reasonable given the premise of the paper. Figures and tables are comprehensive and helpful.

The paper generates the following kinds of data:
1) Bacterial isolates from a marine environment
2) Microbial materials collected on filters from coral and marine environments
3) Sequencing reads from bacterial isolates
4) Sequencing reads from metagenomic samples
5) Assemblies of microbial genomes
6) Custom scripts

As the reviewer I am unable to verify that the assemblies or read data are going to be publicly available. There is a note indicating that the data is being submitted to Genbank but it was not clear what data (reads, assemblies) was submitted.

There is no explicit mention of whether the isolates or filters have been saved or would be available for other researchers. Particularly for the isolates this would be important to mention.

In fairness the scripts are a minor concern but they do not seem to be available. In keeping with true open access these scripts, with a short description of there usage context, should also be made available.

Finally, I would suggest explicitly indicating how much data was generated (reads, base pairs). Was any quality trimming done? It wasn't indicated in the text.

Experimental design

In general the experimental design was excellent and clearly written. Some minor changes, additions, modifications would be suggested as follows:

1) pg 6 line 159-160: It is somewhat ambiguous to say something is selected at random based on color or size. This should be clarified.
2) Pg 8 ln 198-204: Inconsistent referencing of languages/libraries. PERL language is referenced but scripts unavailable while R and Python languages not referenced but libraries are referenced and available.
3) Pg 8 ln 210-214: What operating system/file system was being used? Was this system in a RAID array?
4) Pg 9 ln 255: This reference is incorrect. The paper is in preparation and should not be cited with a publication date unless it has been accepted.
5) Pg 10 ln 274: Python script not available.
6) Pg 11 ln 303: Dinsdale reference not correctly formatted. It is not clear what packages/libraries or tools were used to carry out these statistical analyses. These should be directly referenced or it should be clear from the text that the reference given reference is the primary references for these tools.

One over arching issue with the methods is that it is not always clear whether specific aspects of the work took place on site on the research vessel or at a later time. Since this is the primary goal of the paper (i.e. to demonstrate the ability to do real time on site metagenomics) this is critical to the paper. Additionally it would be good to indicate how long these steps took to carry out since time may be a real factor and would help other researchers plan similar expeditions or scale their effort appropriately.

Validity of the findings

The results are reasonable given the experiements.

Additional comments

Interesting paper definitely addresses a need of the scientific community. I was surprised not to see more suggestions on how to handle the file corruption issues. Using md5 checksums is fine for validation but use of RAID disk arrays or use of solid state hard drives could probably solve most of these problems.

Reviewer 2 ·

Basic reporting

The paper presented intends to show the advantages of in situ Next Generation Sequencing (NGS) for remote locations. The work here presents 26 marine microbial genome, and two metagenomes. On board sequencing could be interesting though it presented some technical difficulties and its not clear what would be the advantage of real time processing against deep freezing and sending the samples to the sequencing facilities.

Experimental design

The experimental design is the result of an expedition to the Southern Line Islands, with samples collected either from coral or algal-surfaces. The nature of this work is exploratory and descriptive, and a proof of concept of field NGS, which overrides the lack of in depth analysis of the (meta)genomic data as well as linking the sequencing data with the phenotype testing, only shown for the serine utilization experiments. Although some of the concepts and methods for making this possible should be clarified prior to the publication. The observations and concerns are stated in the Comments for the Author section.

Validity of the findings

The results of NGS field sequencing are promising, the capability to sequence and analyze large datasets without Internet access as well as servers, HPC clusters or cloud services is interesting. This kind of techniques could be interesting and useful for research groups with no access to large computing infrastructure. And some of the scripts and methodology used should be disclosed in order to test results reproducibility.

Additional comments

P. 7 ~ L 170 What is an appropriate scale of B3 lysis?
P.8 ~ L 207 What is modified from the Ion Torrent pipeline? Is there a chance to document this? I only noticed the MD5 checksum, and the crop into four quadrants. Is there anything else?
P.8 ~ L 226 Could you share your custom Perl script, like in figshare?
P.9 ~ L 235 Do you think you could share your annotation pipeline scripts? This would be of interest for the whole community trying to annotate on locations with poor internet connections or limited computing resources.
P.9. L 249 Did you perform comparisons of your custom annotation against the standard RAST pipeline? This should be included into a summary of results.
P.10 ~ L 270 The e-value is dependent on the database size, could you please state what is the effective database size.
P.10 ~ L 279 Would you share your python script?
P. 17 ~ L 467 The analysis did not demonstrated as stated, it only suggest. Replace demonstrated with suggests.
P. 17 ~ L 468 Change identified their ability to have the predicted potential.
P. 18 ~ L 500 Could you state where you remove centrifugation steps from your procedure in methods?
Is there any chance to compare the results from in situ to frozen samples processed with the regular DNA extraction/sequencing protocols?
P.18 ~ L 508 Could you please describe what where the steps of the reverse engeneering? You were so lucky to have such a hacker on board!

---

## Round 0.2 · accepted · Accept

thanks for the corrections, it is a great paper!!

---

## Author Rebuttal · Round 0.2

**College of Sciences**
San Diego State University
5500 Campanile Drive
San Diego CA 92182-7720
Tel: +1-619 - 594 - 1672
Fax: +1-619 - 594 - 6746
http://www.sdsu.edu
redwards@mail.sdsu.edu

July 16th, 2014

Dear Editors

We thank the reviewers for their generous comments on the manuscript and have have edited the manuscript to address their concerns.

In particular all of the code we wrote is available and I have included multiple links throughout the paper to the appropriate code repositories.

We believe that the manuscript is now suitable for publication in PeerJ.

Dr. Robert A. Edwards
Associate Professor of Computer Science and Biology

On behalf of all authors.

*Reviewer 1 (Douglas Rusch)*

*Basic reporting*

*This paper is clearly written and well organized. The introduction and background are reasonable given the premise of the paper. Figures and tables are comprehensive and helpful.*

*The paper generates the following kinds of data:*
*1) Bacterial isolates from a marine environment*
*2) Microbial materials collected on filters from coral and marine environments*
*3) Sequencing reads from bacterial isolates*
*4) Sequencing reads from metagenomic samples*
*5) Assemblies of microbial genomes*
*6) Custom scripts*

*As the reviewer I am unable to verify that the assemblies or read data are going to be publicly available. There is a note indicating that the data is being submitted to Genbank but it was not clear what data (reads, assemblies) was submitted.*

We have submitted the raw reads to the short read archive and the assembled, annotated sequences to GenBank. We have also made all the annotations available to the guest account in RAST (username guest, password guest). We have added table 3 to the manuscript that includes all the appropriate accession numbers.

*There is no explicit mention of whether the isolates or filters have been saved or would be available for other researchers. Particularly for the isolates this would be important to mention.*

The research permits and import regulations of the US restricted our ability to retain viable bacteria. We retained the isolated DNA and sequence libraries that are not controlled by these permits.

*In fairness the scripts are a minor concern but they do not seem to be available. In keeping with true open access these scripts, with a short description of there usage context, should also be made available.*

This is quite correct. I have included links to all the scripts mentioned in the manuscript.

The multi-phenotype assay plate analysis software is available from [https://github.com/dacuevas/PMAnalyzer](https://github.com/dacuevas/PMAnalyzer) (line 205)

The code to separate sequences by tag is available from [http://edwards-sdsu.cvs.sourceforge.net/viewvc/edwards-sdsu/bioinformatics/bin/split_sequences_by_tag.pl](http://edwards-sdsu.cvs.sourceforge.net/viewvc/edwards-sdsu/bioinformatics/bin/split_sequences_by_tag.pl) (line 240).

The code to assign functions to sequences (annotate the genomes) is a part of the SEED toolkit and is available from [http://biocvs.mcs.anl.gov/viewcvs.cgi/FigKernelScripts/assign_to_dna_using_kmers.pl](http://biocvs.mcs.anl.gov/viewcvs.cgi/FigKernelScripts/assign_to_dna_using_kmers.pl) (line 264)

Other Perl and Python scripts written during the analysis are available from  http://edwards-sdsu.cvs.sourceforge.net/viewvc/edwards-sdsu/bioinformatics/LineIslandsGenomes/

*Finally, I would suggest explicitly indicating how much data was generated (reads, base pairs). Was any quality trimming done? It wasn't indicated in the text.*

Table 2 details the volume of data generated in both reads and base pairs. The only quality trimming was removing sequences that did not match a primer (line 246). We did not remove other sequences prior to assembly. A comment to that effect has been added at that line.

*Experimental design*

*In general the experimental design was excellent and clearly written. Some minor changes, additions, modifications would be suggested as follows:*

*1)      pg 6 line 159-160: It is somewhat ambiguous to say something is selected at random based on color or size. This should be clarified.*

Agreed. I have deleted the words "at random"

*2)      Pg 8 ln 198-204: Inconsistent referencing of languages/libraries. PERL language is referenced but scripts unavailable while R and Python languages not referenced but libraries are referenced and available.*

I have added the statement that all code is available and provided a link to where everything is available from (https://github.com/dacuevas/PMAnalyzer).

*3)      Pg 8 ln 210-214: What operating system/file system was being used? Was this system in a RAID array?*

This is the single hard drive in the Ion PGM (it is a Western Digital harddrive in our machine). I added a clarifying comment on line 214.

*4)      Pg 9 ln 255: This reference is incorrect. The paper is in preparation and should not be cited with a publication date unless it has been accepted.*
Agreed. I have corrected this.

*5)      Pg 10 ln 274: Python script not available.*

This script is available from http://edwards-sdsu.cvs.sourceforge.net/viewvc/edwards-sdsu/bioinformatics/LineIslandsGenomes/  and I have added a comment to that effect in the revised paper line 294)

*6)      Pg 11 ln 303: Dinsdale reference not correctly formatted.*

Corrected

*It is not clear what packages/libraries or tools were used to carry out these statistical analyses. These*

*should be directly referenced or it should be clear from the text that the reference given reference is the primary references for these tools.*

I have added the reference and the package (the R package randomForest).

*One over arching issue with the methods is that it is not always clear whether specific aspects of the work took place on site on the research vessel or at a later time. Since this is the primary goal of the paper (i.e. to demonstrate the ability to do real time on site metagenomics) this is critical to the paper. Additionally it would be good to indicate how long these steps took to carry out since time may be a real factor and would help other researchers plan similar expeditions or scale their effort appropriately.*

I have added a comment to all of the methods identifying which methods were performed on the ship and which were pursued subsequent to our return.

I have also added this statement at line 355 which addresses the timing. Culturing, DNA extraction, library construction, and sequencing took approximately 5 days to complete. The analysis of the sequence chips took approximately 5 hours per quadrant to complete, and the annotation of those sequences took about an hour to complete. As noted below, analysis of the genomes and metagenomes was the most time consuming part of the bioinformatics analysis, and remains an ongoing project.

*Validity of the findings*

*The results are reasonable given the experiments.*

*Comments for the author*

*Interesting paper definitely addresses a need of the scientific community. I was surprised not to see more suggestions on how to handle the file corruption issues. Using md5 checksums is fine for validation but use of RAID disk arrays or use of solid state hard drives could probably solve most of these problems.*

I completely agree. I have added a comment at line 555.
*Reviewer 2*

*Basic reporting*

*The paper presented intends to show the advantages of in situ Next Generation Sequencing (NGS) for remote locations. The work here presents 26 marine microbial genome, and two metagenomes. On board sequencing could be interesting though it presented some technical difficulties and its not clear what would be the advantage of real time processing against deep freezing and sending the samples to the sequencing facilities.*

There are many reasons why researchers will want to use sequencing in remote locations, including the limitations of permitting (in our case), archiving samples, and the ability to perform experiments as soon as the data is generated. We have added this comment to the discussion.

*Experimental design*

*The experimental design is the result of an expedition to the Southern Line Islands, with samples collected either from coral or algal-surfaces. The nature of this work is exploratory and descriptive, and a proof of concept of field NGS, which overrides the lack of in depth analysis of the (meta)genomic data as well as linking the sequencing data with the phenotype testing, only shown for the serine utilization experiments. Although some of the concepts and methods for making this possible should be clarified prior to the publication. The observations and concerns are stated in the Comments for the Author section.*

We are also working (always working) on meta-analyses of metagenomes that come from the Line Islands. We now have a decade of sampling from this unique environment, and a detailed comparison of those samples will be a separate paper.

*Validity of the findings*

*The results of NGS field sequencing are promising, the capability to sequence and analyze large datasets without Internet access as well as servers, HPC clusters or cloud services is interesting. This kind of techniques could be interesting and useful for research groups with no access to large computing infrastructure. And some of the scripts and methodology used should be disclosed in order to test results reproducibility.*

We apologize for this omission. All of the code are currently in public repositories, and I have added detailed links to the software throughout the manuscript. I have detailed these in the response to reviewer #1.

*Comments for the author*

*P. 7 ~ L 170 What is an appropriate scale of B3 lysis?*

I have corrected this to read  An appropriate volume  (200µl for 180µl T1 lysis buffer added, and 400µl for 360µl T1 lysis buffer added) of B3 lysis buffer was added for complete lysis before the lysate was removed from the Sterivex filter for subsequent extraction procedure as described in the manufacturers protocol

*P.8 ~ L 207 What is modified from the Ion Torrent pipeline? Is there a chance to document this? I only noticed the MD5 checksum, and the crop into four quadrants. Is there anything else?*

Those are the only two modifications, although I also ran the commands individually rather than using a  pipeline per se. For clarity, I have included the commands in the revised manuscript.

*P.8 ~ L 226 Could you share your custom Perl script, like in figshare?*

All of the scripts are now linked to their appropriate resources.

*P.9 ~ L 235 Do you think you could share your annotation pipeline scripts? This would be of interest for the whole community trying to annotate on locations with poor internet connections or limited computing resources.*

All of the scripts are now linked to their appropriate resources. I have also included a link to the real time metagenomics site which essentially does the same thing. Note that this requires a kmer-function database as described in Edwards et al. 2012. We are also working on a new version of this where we will release code to build the databases from any protein families and fast ways to annotate metagenomes based on those kmers.

*P.9. L 249 Did you perform comparisons of your custom annotation against the standard RAST pipeline? This should be included into a summary of results.*

No, we did not. We know that the k-mer based annotations tend to over-call the presence of genes (since we sacrifice accuracy for speed). We have compared metagenome annotations to those using BLAST in Edwards et al. 2012.

*P.10 ~ L 270 The e-value is dependent on the database size, could you please state what is the effective database size.*

The metagenomes from 996,778 bp to 117,975,100 bp and we did not concatentate them before doing blast, and so we used expect value cutoffs to account for the difference in size.

*P.10 ~ L 279 Would you share your python script?*

This script is available from  *http://edwards-sdsu.cvs.sourceforge.net/viewvc/edwards-sdsu/bioinformatics/LineIslandsGenomes* and we have added a comment in the text.

*P. 17 ~ L 467 The analysis did not demonstrated as stated, it only suggest. Replace demonstrated with suggests.*

*Corrected.*

*P. 17 ~ L 468 Change identified their ability to have the predicted potential.*

*Corrected.*

*P. 18 ~ L 500 Could you state where you remove centrifugation steps from your procedure in methods? Is there any chance to compare the results from in situ to frozen samples processed with the regular DNA extraction/sequencing protocols?*

Here are the examples where centrifugation was removed from the procedure:
   (i)     Isolated bacterial cells are commonly grown in broth culture and pelleted before DNA extraction. However, in order to avoid centrifugation, individual isolated colonies were grown on agar plate to reach a dense colony prior to DNA isolation.
   (ii)    All centrifugation steps for pelleting ISPs prior to sequencing were done using the centrifuge within the OneTouch Instrument in the bottom of the boat rather than a separate microfugre.

We have edited the text here.

*P.18 ~ L 508 Could you please describe what where the steps of the reverse engeneering? You were so lucky to have such a hacker on board!*

Its not glamorous! Basically, I started by connecting the one touch to my laptop (a System 76 Linux machine) via an Ethernet cable. By inspecting the laptop logs, I saw that the one touch requested an IP address, so then checked to see if an ssh server was running by using telnet to port 22.  Luck! Now all I had to do was figure out the username and password. Based on my work with Ion they tend to use some combination of ion, ionuser, ionadmin, or root as the username, and a similar combination for the password. Success, the combination of root/ionadmin let me into the machine. Then I had to poke around for a while to figure out how the screen is controlled. It uses a piece of software called onetouch … that was a big clue. Since I used X-windows on my laptop I tried running the onetouch command, but the system hard drives on the one-touch are mounted read only. Remounting the hard drive with read/write permissions allowed me to run the onetouch application on my laptop. Although that sounds trivial it was several hours of hacking to get there, meanwhile everyone else was out diving on the reefs collecting samples!

We have replaced the screen but have still not fixed the problem, and now our students use a laptop to control the one touch all the time!

I did not put this detail in the manuscript but could if you would like.